# Open versus Transcutaneous (Ultrasound-Guided and Based on Anatomic Landmarks) Tunneled Venous Access to the Right Internal Jugular Vein in Children: A Prospective Single-Center Study

**DOI:** 10.3390/diseases11040174

**Published:** 2023-11-30

**Authors:** Niki Kouna, George Noutsos, Christina Koufopoulou, Dimitrios Panagopoulos, Antonis Kattamis

**Affiliations:** 12nd Department of Anesthesiology, Attiko University Hospital, 12462 Athens, Greece; nkouna26@gmail.com; 2Pediatric Hospital of Athens, Agia Sophia, 11527 Athens, Greece; georgenoutsos60@gmail.com; 3Department of Anesthesiology, Medical School, National and Kapodistrian University of Athens, Areteio Hospital, 11528 Athens, Greece; chriskoufopoulou@gmail.com; 4Pediatric Oncology, First Department of Pediatrics, University Pediatric Hospital of Athens, Agia Sophia, 11527 Athens, Greece; ankatt@med.uoa.gr

**Keywords:** open surgical technique, percutaneous insertion via external landmarks, ultrasound-guided insertion, thrombosis, infection, catheter malposition, learning curve

## Abstract

Background: The purpose of this study was to compare the immediate and long-term complications that are associated with the utilized techniques for the insertion of indwelling central venous catheters, that is the open surgical technique, the ultrasound-guided technique, and the transcutaneous technique based on external anatomical landmarks in the right internal jugular vein, to a pediatric population. Methods: This was a prospective randomized trial based on a pediatric patient population under 16 years of age of a tertiary pediatric-oncological hospital. The procedure was performed by a medical team with varying experience regarding the percutaneous and open insertion methods. We studied the outcome of our procedure, based on the immediate and delayed complication rate, as well as the needed time in order to complete the procedure and mean duration of line use. Results: The patients that were inserted in our protocol were divided into three subgroups based on the selected technique for the insertion of the central venous catheter. A total number of 88 insertions (25.4%) (out of 346) were based on the technique that was using external anatomical landmarks, 121 insertions were based on the ultrasound-guided transcutaneous technique (34.9%), whereas in 137 cases (39.5%) the open surgical technique was preferred. All cases that were related to catheter re-insertion were excluded from our study. We performed a statistical analysis regarding the catheter dwell time between the three subgroups of patients and no significant difference was recorded. Moreover, the development of thrombosis was investigated, and we noted that a higher percentage of this complication was related to the transcutaneous external landmark and open surgical technique. Also, the incidence of infection was taken into consideration, which manifested an increased incidence when the transcutaneous technique based on external landmarks was used. Conclusions: Ultrasound-guided percutaneous insertion was considered to be a safe and effective technique for the insertion of central venous catheters. Our study also demonstrated a decrease in operating times when performed by operators with increasing expertise, increased preservation of the diameter of the venous lumen, and no increase in complication rates when the ultrasound-guided technique was selected.

## 1. Introduction

It has been reported that the incidence of occlusion of the internal jugular vein after the insertion of a central venous catheter is in the range of 33% in the pediatric population when the open surgical technique is used [1]. On the contrary, the relevant percentage is 3%, in cases where a transcutaneous approach has been selected [1]. Nevertheless, the open surgical technique is the most commonly utilized technique worldwide, whereas the studies that are centered on the use of transcutaneous techniques in children are exceedingly rare [2,3]. To the best of our knowledge, there are only three published studies that attempted a comparison between the open surgical and the transcutaneous technique for the insertion of central venous lines in the pediatric population. However, they share in common several methodological disadvantages· all of them are retrospective in nature, they are not well-organized, and due to methodological errors, they present conflicting results [4,5,6].

Basford et al. [4] mentioned that when the central venous catheters were inserted via ultrasound guidance, the reported incidence of complications was lower compared with the cases where an open surgical technique was preferred. A drawback of this survey is based on the fact that under the term ‘open surgical insertion’ were also included catheters that were inserted via an external anatomical landmark technique. Also, this survey included all cases of catheter insertion, irrespective of the recipient vessel, i.e., the subclavian vein and the internal and external jugular vein. Avanzini et al. [5] published their results that were derived from a series of data based on 129 open central venous catheter insertions and 66 insertions via a transcutaneous, ultrasound-guided technique. A drawback of their survey is that they have not reported the results of their statistical analysis, centered on the comparison of the two different insertion techniques. In addition, they collectively analyzed their data, and they did not subdivide them based on the selected vessel for insertion of the catheter. This means that their analysis was based on data that involved the internal and external jugular vein, as well as other recipient vessels which are not reported, and not separately for each recipient vessel. 

The administration of chemotherapy, total parenteral nutrition, or replacement of blood factors in children is based on the utilization of tunneled Hickman catheters along with implantable venous access devices. The most commonly utilized technique is the open surgical approach, which is traditionally performed through a venotomy. An alternative method is based on a percutaneous procedure which is based on external anatomical landmarks or on ultrasound guidance. Based on recent recordings, the insertion of central venous catheters via a percutaneous access under ultrasound guidance is increasingly used nowadays. Moreover, according to published studies, this is currently recommended as the procedure of choice in order to achieve central venous access [7,8]. Apart from that, a recently published study based on a population of 500 children has demonstrated that the utilization of ultrasound-guided percutaneous access is considered as a safe alternative to the open surgical technique and is accompanied with minimal complications [2]. A review of the published data derived from recent bibliographic reports suggest that the ultrasound-guided insertion of central venous lines has proven to be a safe therapeutic option, minimizing complications associated with blind needle punctures [9,10,11]. Although this strategy has been established as a standard of practice for the adult patient population, the same is far from a universally accepted method to their pediatric counterpart. To the best of our knowledge, there are a lot of reports comparing the ultrasound-guided percutaneous access to the percutaneous insertion via external anatomical landmarks. On the contrary, studies comparing the aforementioned technique with the open surgical access are lacking in the literature. After performing bibliographic research, we recorded only one study, which was based on a randomized trial comparing open versus ultrasound-guided percutaneous catheter insertion in a pediatric cohort of patients [7].

Taking into consideration all these limitations of previous studies, we planned this prospective study, aiming to achieve a comparison of the immediate and delayed complications that are associated with the open surgical technique, the ultrasound-guided technique and the transcutaneous technique based on external anatomical landmarks in the right internal jugular vein of a pediatric population. It has been recorded that the incidence of occlusion of the internal jugular vein following the insertion of a central venous catheter via a transcutaneous technique in the pediatric patient group is considerably lower when it is compared with the relevant percentage that is related with the utilization of the open surgical technique. 

The aim of the present study is to compare the immediate and long-term (delayed) complications that are associated with the utilization of the transcutaneous technique based on external anatomical landmarks, the ultrasound-guided transcutaneous technique, as well as the open surgical approach. Under the term ‘immediate complications’ are collectively included these that are intimately related with the operative procedure, as well as those that took place at an interval of 30 days after the catheter implantation. On the contrary, the delayed complications are those that are recorded at least 30 days after the procedure. Thrombosis of the internal jugular vein is included under this term. 

## 2. Materials and Methods

The insertion of the central venous catheters was performed between 2016 and 2022, whereas the removal of the relevant catheters was performed in the same time period. The patient population that was incorporated in our study included patients that were nursed at the pediatric hematology and oncology departments of our hospital, which is a tertiary pediatric hospital. Our patients age ranged from 0 to 18 years old (Table 1, Table 2 and Table 3; Figure 1, Figure 2 and Figure 3) and included patients who necessitated insertion of long-staying central venous catheters.

Patients that were bone marrow recipients, as well as those that necessitated long-term chemotherapy constituted the vast majority of the individuals that were candidates for insertion of indwelling venous catheters (Figure 4).

The vessel that was universally selected for catheterization was the internal jugular vein and this is a feature that was not encountered in other relevant studies (Table 3). The utilized central venous lines were implantable, open-ended, silicone-made, Hickman-type catheters. They were divided into one (single) lumen (diameter 4.2, 6.6, 9.6 French), as well as double lumen (7, 9 French) (Table 4 and Table 5, Figure 5).

All cases that involved insertion of catheters to other venous branches (i.e., external jugular vein, subclavian vein) were automatically excluded from our survey. 

The following is a detailed description of the insertion procedure, including the open surgical technique, the ultrasound-guided technique, and the transcutaneous technique based on external anatomical landmarks (Table 6 and Table 7, Figure 4).

We have also recorded all complications that were associated with catheter insertions· a relevant table depicting a detailed description of them is following (Table 8).

All procedures were performed under general anesthesia. Children were positioned supine with a roll under the shoulder and neck rotated slightly to side opposite to site of insertion. The equipment used was Sonosite M-Turbo ultrasound (Fujifilm Sonosite, Inc., Bothell, WA, USA) and transducer was SLAx hockey stick transducer. 

### 2.1. Percutaneous Access

The initial needle puncture was performed under ultrasound guidance. The needle was passed perpendicular to transducer axis and in line with IJV. The vein was punctured under vision and blood aspirated to confirm that the lumen was entered. Bard percutaneous (Bard Access Systems, Inc., Salt Lake City, UT, USA) access kits were used for insertion of double or single lumen catheters depending on age, weight, and indication for insertion. Vortex kits (Angiodynamics, Latham, NY, USA) were used for insertion of implantable vascular access device. In infants and children under 10 kg, a conversion kit (Arrow International Inc., Brooklyn, OH, USA) was used to gain access into the vein. The exit site was over the pectoral region. The catheter was cut to length with an image intensifier (II) prior to insertion to position at junction of superior vena cava and right atrium. The position and function of the line was confirmed by II prior to closure of skin wounds.

### 2.2. Open Access

IJV was accessed through a low-neck incision along skin crease over the sternomastoid muscle. IJV was exposed through the triangle between the two heads of SM muscle and looped. The exit site was chosen over pectoral region and catheter was tunneled through. The catheter was cut to length at the level of the sternal angle to position it at the level of junction of superior vena cava and right atrium junction. The venotomy was made and the catheter inserted. The position was confirmed with II and 6-0 prolene purse string suture applied to venotomy to achieve hemostasis. The catheter function was checked and the wound closed.

### 2.3. Insertion of Central Venous Line via External Anatomical Landmarks

General steps include:

Place the patient supine with their head turned gently to the left. This central venous access technique via the right jugular vein will be successful no matter how much or how little the head can or cannot be turned. 

The practitioner’s left ring finger is placed in the patient’s sternal notch, and the adjoining middle and index fingertips are brought together such that they are in the midline over the trachea.

This group of fingers is then rolled over the trachea and down into the space between the trachea and medial head of the medial sternocleidomastoid muscle. The pads of the three fingers must stay in contact with the trachea. When performed properly, the left ring finger is now in contact with the sternoclavicular joint. The medial head of the sternocleidomastoid muscle is bunched into a mound lateral to the fingers. In the middle of the mound and at the level of the left index finger, insert the finder needle at a 45-degree angle to the skin, aiming toward the patient’s ipsilateral nipple with gentle suction applied to the syringe. 

Once the vein has been entered, and venous blood is positively identified in the syringe, the larger introducer needle is inserted directly over the top of the smaller finder needle at the same angle and direction as the finder needle. We then proceed to place a catheter with the Seldinger technique.

There were no inclusion and exclusion criteria for each group. On the contrary, the selection of the insertion technique was based on a randomization procedure.

As a continuation of our previous statement, we would like to mention that the open surgical technique was not conducted because of the failure of the other two procedures.

It is crucial to shortly mention and describe an anatomical explanation of children and adult differences. Children’s cranial and spinal anatomy undergo many changes, from the presence and disappearance of the fontanels, the presence and closure of cranial sutures, the thickness and pliability of the cranium, anatomy of the vertebra, and the maturity of the cervical ligaments and muscles. Moreover, their systemic anatomy changes over time. The head is relatively large in young children, relative to the rest of their body, the airway is easily compromised, the chest is poorly protected, and the abdominal organs are large. 

A total of 443 patients were included in our survey and the vessel that was selected for catheterization was the internal jugular vein. The right jugular vein was preferred in the majority of these patients (346 patients, 78.1%), whereas the left jugular vein was involved in only 97 patients (21.9%). Our study was centered only on those patients for whom the right internal jugular vein was selected for catheterization.

The patients that were inserted in our protocol were divided into three subgroups based on the selected technique for the insertion of the central venous catheter.

Group A incorporated children aged under 16 years, to whom an indwelling central venous catheter was inserted transcutaneous into the internal jugular vein via ultrasound guidance (US-guided percutaneous Seldinger technique).

Group B included children aged under 16 years, to whom an indwelling central venous catheter was inserted transcutaneous into the internal jugular vein via external anatomical landmarks (percutaneous Seldinger landmark technique).

Group C was based on children aged under 16 years, to whom an indwelling central venous catheter was inserted via the open surgical technique (cutdown technique). This group was used as a control group, as this method constitutes the most commonly used technique worldwide for the insertion of central venous lines. In order to avoid bias, we performed matching of the three groups, based on the age of all the participants. All cases that involve insertions of the central venous catheters to other vessels than the internal jugular vein (i.e., external jugular vein, subclavian vein) were not included in our study. All our patients were followed-up for a period of one year after catheter insertion. 

After the insertion of the central venous line, we recorded all immediate and delayed complications related to the insertion of the catheter. Under the term immediate complications, we included all peri-operative ones and those that were evident within 30 days after the insertion. All cases to which the insertion was accomplished on the opposite side were considered as failure of insertion and were excluded from our study.

A total number of 88 insertions (25.4%) (out of 346) were based on the technique that was using external anatomical landmarks, 121 insertions were based on the ultrasound-guided transcutaneous technique (34.9%), whereas in 137 cases (39.5%) the open surgical technique was preferred. All cases that were related to catheter re-insertion were excluded from our study and, as a result, the relevant final percentages were 26.5%, 32.5%, and 41%, respectively. Besides that, we also excluded all cases of patients that the insertion of the catheter was associated with failure. After that, the total number of catheter insertions was 258, that is 62 cases for the technique based only on external anatomical landmarks, 86 cases for the ultrasound-guided technique, and 110 cases for the open surgical technique. 

After that, the participants were also subdivided based on their age and we recorded our preference to any specific technique, based on that criterion. It seems that were the referring patients were between 0 and 1 years old, the ultrasound-guided transcutaneous technique was our first choice, whereas in the 1–5 years old group, the open surgical technique was most commonly utilized. This was also the case for patients that were older than 5 years of age. 

The criteria for participation to our protocol were fulfilled by 443 patients. A total of 346 insertions to the right internal jugular vein were performed. Among them, 268 cases involved cases who were operated on for the first time. All patients that underwent a re-insertion of the central venous line were excluded from our study.

The optimum position of the terminal point of the central venous catheter was considered to be the junction of the internal jugular vein with the right atrium and it was verified via an intra-operative antero-posterior chest X-ray. Every other terminal point was regarded as erroneous or suboptimal and these patients were not included in our study. In order to avoid bias (false differences between subgroups) to our results, due to age differences between them, we performed matching of three groups related to the age of the participants. 

For the present study, we attempted to perform a comparison of the three aforementioned techniques for the insertion of central venous lines. We investigated the relative incidence of complications among these groups, and we attempted a statistical analysis of the relevant results.

A total number of 14 patients (out of 268) were excluded from the continuation of our study because of inappropriate/suboptimum position of the end of the central venous catheter. Two of these patients participated in the group of children to whom the central venous line was inserted via the external anatomic landmark approach and three of them were under the team that the insertion was guided via a transcutaneous ultrasound. At last, nine patients belonged to the group that the open surgical technique was performed. 

Apart from that, another group of 5 patients were excluded from our survey, as they continued their therapy to another medical center. Moreover, 12 patients died due to disease-related complications. 

### 2.4. Statistical Analysis

Quantitative variables are expressed as the mean ± standard deviation (SD) and interquartile ranges. The qualitative variables are expressed as the absolute (N) and relative (%) frequencies. The different ratios among multiple groups were compared using Pearson’s chi-square or Fisher’s exact test. The quantitative variables among multiple groups were compared using parametric analysis of variance (ANOVA) or the non-parametric Kruskal–Wallis test. The significance levels were two-sided and the statistical significance was set at 0.05. All analyses were performed using GraphPad Prism v10 software.

## 3. Results

Our study included 142 male patients (53.4%), as well as 126 (46.6%) female patients. The mean age of the participants was 7.9 ± 5.1 years. More specifically, the relevant age of the patients for whom the transcutaneous technique based on external landmarks was preferred was 8.7 ± 5.1 years, whereas the relevant mean age for those the open surgical technique was preferred, was 7.3 ± 5.1 years. Our statistical analysis was unable to verify the existence of any statistically significant difference between these three groups of patients, when their mean age was analyzed. The mean body weight of all the participants was 29.3 ± 18.1 kg. 

The underlying disease of all the aforementioned patients were malignancies (hematological diseases or other oncological entities). These patients received chemotherapy or they were candidates for bone marrow transplantation. The most commonly recorded disease was acute lymphoblastic leukemia, diagnosed in 84 patients (32.55%). The second most common diagnosis was sarcoma (39 patients, 15.11%). In decreasing order of frequency, the patients’ diagnoses were brain tumors (24 patients), lymphoma (23 patients), acute myeloblastic leukemia (22 patients), and solid organ tumors (21 patients). The number of candidates for bone marrow transplantation that were suffering from anemia was 30. 

When the underlying indication for the insertion of a central venous catheter was considered, the administration of chemotherapy was the most commonly reported (212 patients, or 76.5%). Bone marrow transplantation was performed in 65 patients (23.5%). 

We attempted to investigate if there was a predilection for a specific insertion technique, in association with the underlying disease process. We mentioned that the open surgical procedure was the most commonly used insertion technique, irrespective of the disease process (hematological disease or tumors of solid organs). More precisely, the open surgical technique was selected in 58/129 of patients with hematologic malignancies, whereas the same technique was adopted in 10/21 patients, harboring a solid tumor organ.

Re-insertion of a central venous catheter was performed in 78 out of 346 cases in order to facilitate the completion of therapy. The underlying cause was an accidental removal, catheter rupture, infection, malfunction, or thrombosis. Apart from that, catheter re-insertion was attempted in cases that we had diagnosed recurrence of malignancy in order to proceed to bone marrow transplantation. All these cases were excluded from our survey. A total of nine attempts (out of 78), were unsuccessful (11.5%). The success rate was 96.3%, taking into consideration the total number of 268 first-time insertions. 

Another parameter that was recorded is related to the operative time for the insertion of the central venous catheter. These data were collected for all patients, regardless of the utilized insertion technique. Our data state that the mean operative time in cases that the transcutaneous external landmark technique was selected was 49.1 min, whereas the relevant mean time was 43.6 min when the ultrasound-guided technique was selected. When we adopted the open surgical technique, the mean time was 55.3 min.

### 3.1. Immediate and Long-Term Complications

The total incidence of inappropriate position of the tip of the central venous catheter was 5.42% (Table 9), when all attempts at central venous catheter insertion to the right internal jugular vein were taken into consideration. 

More precisely, when the selected insertion technique was taken into consideration, the relevant incidence was 3.23% when the transcutaneous external anatomical landmarks technique was utilized, whereas the relevant incidences were 3.48% and 8.18%, when the insertion technique was the ultrasound-guided and the open surgical technique, respectively. The total number of recorded complications was 25 out of 258 insertions (9.30%) (Table 10 and Table 11).

The greatest incidence of complications was related to the transcutaneous anatomic landmark technique (24.19%), followed by the ultrasound-guided technique (12.79%) and finally, the open surgical technique (11.81%). The most serious complications were rupture of the carotid artery (5.81%), along with inappropriate position of the tip of the central venous catheter (5.42%). When the selected technique was the transcutaneous external landmark technique, rupture of the carotid artery was the most commonly recorded complication (12.90%), whereas the inappropriate position of the catheter tip was most commonly correlated with the open surgical technique (8.18%). Less commonly recorded complications included the development of hemothorax (1.16%), hemorrhage (0.77%) pneumothorax (0.77%) suture vessel ligation (0.38%), hemopericardium (0.38%), as well as pseudoaneurysm formation (0.38%). 

As we already mentioned, inappropriate position of the tip of the central venous catheter was a relatively common complication to our patient series. The most common erroneous position of the distal end of the catheter was within the right subclavian vein (6 out of 14 cases, 42.85%). This was followed by the short length of the inserted central venous catheter (4/14, 28.57%). Other relevant reported erroneous positions of the distal catheter tip were associated with a disproportionately long length of the inserted catheter and cases where the catheter tip was imagined within the azygos vein, mediastinum, or the left internal jugular vein (7.14% of all cases with suboptimum termination of the catheter tip) (Table 12, Figure 2). 

At the end of follow-up of our study, 258 central venous catheters were inserted and 31 of these patients were harboring this central catheter (12.01%). Based on that, we recorded a total number of 227 catheter removals due to a variety of reasons. These included cases of elective removal due to completion of therapy, as well as cases of premature removal (non-elective) due to serious complications (Table 13). 

More precisely, from a total amount of 227 removals, 163 cases (71.8%) were related to elective catheter removal due to completion of the appropriate therapy. The ultrasound-guided technique was associated with the highest percentage of completion of therapy, that is 61 out of 77 cases, which corresponds to a success rate of 79.22%. The relevant percentages associated with the transcutaneous technique based on anatomical landmarks were 41 out of 58 patients (70.68%) and 61 out of 92 patients (66.30%), when the open surgical technique was considered (Table 14).

When the catheter dwell time was compared between the three selected methods for catheter insertion, no statistically significant difference was registered among all compared techniques. The total catheter mean dwell time was recorded in the range of 236 days. 

There were cases that non-elective catheter removal was performed, mainly due to the existence of serious complications. Infection, catheter malfunction, and venous thrombosis were the most commonly encountered relevant complications. Based on our data, infection seemed to be the most common reason of premature catheter removal, and these cases were predominantly related with the external landmark and the open surgical technique. The relevant percentages were 18.96% and 15.21%. Regarding the development of thrombosis, the higher percentage of this complication was related with the transcutaneous external landmark and open surgical technique, that is 8.62% and 7.60%, respectively. We would like to mention that no case of venous thrombosis was associated with the ultrasound-guided technique. Malfunction of the catheter was recorded in 13 cases in total, and five of them were associated with the open surgical technique. 

Our study also recorded the relevant incidence of thrombosis, which was 5.28%. According to the specific subpopulations of patients, the relevant incidence was 8.62% when the patients of the transcutaneous external landmark technique were taken into consideration, whereas the relevant percentage was 7.60% when the open surgical technique was analyzed. A point that deserves special mention is related to the fact that no cases of thrombosis were recorded in the group of patients where the ultrasound-guided technique was utilized. The cases of venous thrombosis were diagnosed in patients that were either harboring a central venous catheter or the catheter has been previously removed due to the reasons already mentioned, namely those associated with the removal of central venous catheters.

Infection was another factor that was intimately related to premature central venous catheter removal. The reported incidence of premature catheter removal due to this complication was 14.10% (32 out of 227 cases) in our series. When each technique was considered in isolation, we mentioned that the incidence of infection was increased when the transcutaneous technique based on external landmarks was used (18.96%). On the contrary, the relevant incidences were 15.21% and 9.09% when the open surgical technique and the ultrasound-guided technique were utilized.

### 3.2. Learning Curve

Another important aspect of our survey was intimately related to the learning curve related with the ultrasound-guided transcutaneous technique during our study. It seems that there was a trend toward an increasing utilization of this method, which reached its maximum during the last year of the study. More precisely, at the beginning of our protocol, it was utilized in about 20% of our participants, whereas this percentage was subsequently increased to >42% at the end of our survey. This means that with time, our personnel gained significantly more experience and confidence with this method.

## 4. Discussion

The insertion of central venous lines constitutes one of the most commonly performed operations by an experienced medical team at a tertiary pediatric hospital. These indwelling central venous catheters are important tools in the management of pediatric-oncological patients suffering from hematological malignancies or who are candidates for bone marrow transplantation. Regardless of the insertion technique, the dwell of these catheters is related to complications which result in their premature removal.

To the best of our knowledge, our bibliographic research revealed very few comparative studies that included all three aforementioned insertion techniques. The vast majority of them were retrospective in nature, not well-organized, and their results presented conflicting evidence. Our study attempted to compare all these insertion techniques of indwelling central venous lines, in terms of safety and efficacy. Several relevant studies have been performed which have utilized for catheterization venous branches of intermediate internal vessel diameter, located in the neck, extremities, and trunk. A recent study attempted a comparison between the ultrasound-guided technique along with the open surgical technique in cases of catheterization of the left and right internal jugular vein [7]. A factor that differentiates our survey from previously published series is based on the fact that the selected vessel for catheterization was strictly the right internal jugular vein. The catheterization of that vessel is preferred due to a variety of reasons. There is lack of any proximity with the great thoracic duct, relatively straight trajectory in association with the right atrium, and the ipsilateral pleural dome lays to a lower level. The trajectory of the vessels is straightforward in relationship with the superior vena cava, thus reducing the risk of inappropriate positioning of the tip of the central catheter. Whenever the catheterization of this vessel is not feasible, the left internal jugular vein is our next option. However, this catheterization is more commonly associated with failure, due to anatomical differentiations related to the diameter of the lumen of the vessel, as well as due to the different anatomical relationship with the common carotid artery. Compared with its right counterpart, the left internal jugular vein is longer with a more horizontal course, thus it terminates with a more acute angle to the superior vena cava. We must take into consideration the fact that the pediatric patients that suffer from malignancies present unique difficulties due to the shorter length and relatively limited internal diameter of their venous vessels, which hinder their catheterization. The chronic nature of their malignancies along with the difficulties associated with their feeding often result in the insertion of indwelling venous catheters early during their disease course. This is unavoidably associated with an increasing frequency of multiple and repeated attempts of venous catheterization. 

A study was published by Soundappan et al. [7] based on a group of 55 patients to whom a central venous catheter was inserted via the ultrasound-guided technique. According to them, multiple venous punctures were needed in 22% of their participants but there was no referral that an open surgical technique was needed as a salvage procedure. A subgroup of patients underwent insertion of the central venous catheter via the use of the transcutaneous technique based on external anatomical landmarks. When these patients were studied, more than two punctures were recorded in 38.70% of them. On the contrary, in the subgroup of patients that underwent an ultrasound-guided technique, only 3.48% necessitated more than two punctures. Ultrasound guidance seems to eliminate injuries to the vessel wall, along with the nearby tissues; moreover, it reduces the attempts of vessel puncture when compared with cases where a blind puncture is performed. An advantage associated with ultrasound guidance is that it can reveal the possible anatomical variations, along with the exact depth of the vessel of interest. Monitoring continuously the tip of our needle during vessel puncture enables us to imagine exactly (real time) the position of its tip, until it lies definitively within the lumen of the vessel. This avoids the possibility that the posterior venous vessel wall or the nearby artery are inadvertently punctured. 

A recent study revealed that the percentages of successful venous catheter placements reached 100% in cases where the open surgical technique was selected, in comparison with a success rate of 90.3% when Seldinger technique was performed [12]. Successful venous catheterization in the range of 99–100% has been referred to in studies based on pediatric or adult patients where ultrasound guidance was utilized. This result was confirmed by other similar studies, which were based on an ultrasound-guided technique for venous catheterization [7]. According to those studies, the relevant success rate was 87.32% when the insertion was based only on external anatomical landmarks, whereas the ultrasound guidance was accompanied by a 98.85% success rate. The relevant percentage reached 100% when the open surgical technique was selected. 

It seems that the transcutaneous technique based only on external anatomical landmarks is intimately associated with an increased frequency of failure in cases of attempted central venous line insertion, whereas ultrasound guidance eliminates this possibility. More precisely, the previously referred study mentioned only one case of (~1%) among the group of patients that underwent an ultrasound-guided insertion. Moreover, this was recorded during the initial stages of obtaining experience with that method. Studies that were performed on larger populations composed of adult patients refer similar results, which can be achieved after gaining enough experience [2]. 

A recent study [7] revealed that the percentage of immediate complications was 14% in the group of patients that the ultrasound-guided technique was selected. On the contrary, the relevant percentage of complications in the subpopulation of patients who underwent catheter insertion via an open surgical technique was 8%. When the transcutaneous technique based only on external anatomical landmarks was preferred, the percentage of complications at the time of puncture range between 10 and 20% [2].

Soundappan et al. [7] reported the relevant percentages of immediate (mechanical) complications related with each one of the three insertion techniques. They were 19% when the anatomic landmark method was utilized, whereas when the ultrasound-guided technique was preferred, the percentage was 12.7%. Finally, the relevant percentage was 11.89% when the open surgical method was evaluated. 

Soundappan et al. [7] published their results concerning the inappropriate position of the distal end of the inserted venous catheter, in association with the utilized insertion method. They have not mentioned any such cases when the ultrasound-guided technique was performed, whereas a failure rate of 3.7% was recorded in association with the open surgical technique. 

According to our data, the incidence of inappropriate ending of the distal venous catheter tip appears to be increased in cases where the open surgical technique was performed, a finding that is in accordance with other previously published data. 

To the best of our knowledge, there is lack of data regarding the repositioning of an erroneously inserted venous catheter in the pediatric patient population. Relevant research, based on insertion of central catheters in 187 neonates [13], postulated that an automated correction can be anticipated within 24 h after insertion. 

As we already mentioned, the chest X-ray remains the gold-standard technique for the verification of the positioning of the end of the catheter. Nevertheless, there are enough cases of erroneously inserted catheters where this imaging modality yields a not definitive result. Concerning the antero-posterior chest X-ray, the bi-directional imagination of the domes that are neighboring the vessel of interest poses an obstacle regarding the precise determination of the tip of the venous catheter. In all cases that there is suspicion of suboptimal ending of the catheter, a lateral chest X-ray should be performed. This enables us to determine the precise location of the catheter tip at the anterior, middle, or posterior mediastinum. 

The precise positioning of the tip of the venous catheter is considered as an important risk factor for the development of thrombosis related to central venous catheter insertion (CVC). The prevalence of CVC is higher in patients to whom the tip of the catheter was located within the anonymous vein or within the proximal part of the superior vena cava, compared with those cases that the ending was located within the terminal part of the superior vena cava, at its junction with the right atrium. It has been reported that the risk for the development of thrombosis was 2.6 times greater in cases that the catheter was terminating within the superior vena cava compared with cases where the tip was terminating within the right atrium [14]. 

According to our data, the mean operation time (Figure 5) regarding the insertion of a central venous catheter was 49.1 min when the transcutaneous technique based on external anatomical landmarks was utilized. 

The relevant time needed was measured as 43.6 min when the ultrasound-guided technique was used. Moreover, the relevant time was 55.3 min when the open surgical technique was the procedure of choice. A relevant study [7] mentioned that the mean duration of catheter dwell was 323.3 days in cases the external anatomical landmark technique was selected, whereas when the open surgical technique was performed, the relevant mean duration of catheter dwell was 278.3 days. This study included a population of patients that received total parenteral nutrition, as well as patients that were receiving coagulation factors. 

Additional parameters that were studied with our survey included the type of therapy that was admitted, the recipient venous lumen and the learning curve of the participants to our protocol. Relevant results are depicted on Figure 6, Figure 7 and Figure 8.

Based on our results, we calculated that the mean duration of time regarding dwell of the central venous catheter within the internal jugular vein (from their insertion until the completion of the therapeutic regimen) was 236.0 ± 230.9 days. More precisely, when the external anatomical landmark technique was used, the relevant value was 238.3 ± 235.7, whereas when the ultrasound-guided technique was selected, the mean duration of catheter stay was 242.2 ± 233.8 days. Finally, in cases where the open surgical method was selected, this value was 229.5 ± 227.3 days (*p* > 0.05). Nevertheless, we must mention that our survey included patients with hematologic malignancies, oncologic patients, and children that were candidates for bone marrow transplantation. On the contrary, other relevant studies included patients that were under parenteral nutrition or were receiving blood transfusion products. These patients share in common that their therapy lasts for a longer period of time.

Soundappan [7] reported that the prevalence of central venous catheter-associated infection was 12.5% in the subgroup of patients that underwent an ultrasound-guided insertion and 11.3% when the open surgical technique subgroup was taken into consideration.

The infections that are intimately related with the central venous lines constitute a serious complication which can potentially result to premature removal of the offending line. Cavanna [15] published a study which was dedicated to the insertion of central venous lines to an adult patient population suffering from tumors of the solid organs as well as from hematological malignancies. They referred to an incidence of infection in the range of 9.9%, whereas 2.9% of them necessitated non-elective removal due to infection or thrombosis. Mirro et al. published a study which attempted to compare the transcutaneous insertion of a central catheter based on external anatomical landmarks with the open surgical technique. This was performed on the basis of a patient population composed of 310 pediatric oncological patients. They supported that the transcutaneously inserted catheters were less prone to failure and were related with a lesser incidence of infection compared with their counterparts that were inserted via the open surgical technique. According to Soundappan et al. [7], the incidence of this central venous catheter infection was in the range of 10% for both groups, and this percentage was comparable with results that were published, based on previous relevant studies. 

Soundappan et al. [7] has reported a study that revealed malfunction of the central venous catheters in the range of 1.7%, based on the results of the ultrasound-guided technique. Moreover, Avanzini et al. [5] referred to a malfunction rate in the range of 3.1%, based on the total number of their patients. Martynov et al. [16] stated that the relevant percentage, according to his data, was 3.2%, considering patients that belong to the open surgical insertion technique. Previtali et al. [17] referred to an incidence of malfunction in the range of 9.7%.

According to our survey, the incidence of premature removal of the central venous lines, due to malfunction, was estimated to be 5.72%. Our data support that it was increased (namely, 6.89%) when the external anatomic landmark technique was taken into consideration, whereas the relevant percentage was 5.42% in the subpopulation of patients that underwent an open surgical insertion. The percentage of such premature removals was even lower (5.19%) when patients that underwent an ultrasound-guided insertion of the central catheter were considered. 

Barnacle et al. reported that the incidence of venous obstruction in cases that underwent an ultrasound-guided insertion was less than 3%. Köksoy et al. [18] reported an incidence of venous occlusion in the range of 40%. Among them, 67% were suffering from complete occlusion after a catheter insertion through an external anatomic landmark technique. They hypothesized that this was intimately related to the need for multiple needle punctures of the venous wall. Maizel et al. [19] presented their own results, and they reported an incidence of venous obstruction in the range of 3%, based on patients that underwent an ultrasound-guided insertion. On the contrary, the relevant percentage was 25% in the subpopulation of patients that underwent an open surgical insertion, and 15–40% when patients that underwent a venous catheter insertion via external anatomic landmarks were considered. 

According to our data, the percentage of patients that developed thrombosis of the right internal jugular vein following the removal of the central venous catheter was 5.28%. The relevant percentage was 8.62% when patients that underwent insertion of the catheter via a technique that was based on an external anatomic landmark technique were considered, whereas the relevant percentage was 7.60% when we took into consideration patients that underwent an open surgical technique for the insertion. On the contrary, no cases of thrombosis were reported in the subgroup of patients for whom the ultrasound-guided technique was selected.

The development of thrombosis could potentially result to total obliteration of the lumen of the involved vessel. This could be a disastrous complication in the subgroup of patients that are suffering from chronic disease processes and long-term venous access is mandatory. A recent, relevant study [7] revealed that the reduction in the internal diameter of the involved vessel was considerably greater when the open surgical technique was used, in comparison with the ultrasound-guided insertion technique. 

According to the results of Soundappanetal [7], no case of occlusion of the venous lumen was reported after the removal of the central venous catheter. Nevertheless, in four patients they recognized, at the time of catheter removal, the presence of thrombus at the site of insertion of the central venous catheter into the lumen of the vessel. Moreover, reduction in the internal diameter of the offending vessel at the site of insertion of the central venous catheter was observed in one patient. 

Another advantage of the ultrasound-guided technique is that it enables re-insertion of the central venous catheter to the same vessel, in case it is mandatory. It seems that the occlusion of the lumen of the recipient vessel is less likely in cases where the ultrasound-guided technique is utilized, whereas it seems to be considerably higher in cases that the open surgical technique is selected. The underlying cause that explains this differentiation relates to the venotomy along with the suturing of the vessel wall that are inherent steps of the surgical technique. All these manipulations injure the surrounding tissues along with the vessel wall, which renders catheter re-insertion more difficult and increases the risk of vessel thrombosis. In cases where re-insertion of the venous line to the same recipient vessel is necessary, it is recommended that before catheterization, imagining of the vessel via ultrasound should be performed, to rule out the presence of thrombosis. The development of thrombosis, along with stenosis of the internal diameter of the lumen of the vessel in cases that the surgical technique was performed, is intimately related with manipulation of the offending vessel as well as with the use of sutures in order to achieve regional hemostasis after venotomy [7].

The imaging of the recipient vessel with relevant accuracy before and after its catheterization may be responsible for the reduced number of attempted vessel punctures. In turn, this can provide a reasonable explanation for the decreased prevalence of venous thrombosis that was apparent to our study, whenever the ultrasound-guided technique was performed.

The association between venous infection and thrombosis seems to be bidirectional. The development of thrombi that are anatomically related with the central venous catheter favor the attachment of micro-organisms, facilitating the development of bacteremia. Moreover, the development of infection that is related with an indwelling central venous catheter is capable of causing an inflammatory reaction, the ultimate result of which is the development of coagulation disorders as well as its denser adherence to the lumen of the vessel wall. 

When an infection that is causally related with an indwelling central venous catheter is established, the possibility for the development of an associated thrombosis is increased. According to our data, a patient group which developed an infection related with thrombosis, shared an increased incidence of secondary development of a clinically significant thrombosis, in the rate of 44% (11 out of 25 patients). On the contrary, in patients without infection, the relevant prevalence of secondary clinical thrombosis is in the range of 3% (2 out of 80 patients). There is a linear association between the severity of the infection and the possibility of thrombus formation. 

Based on the data of our survey, it seems that the prevalence of complications was higher when compared with results of literature research. The explanation of this difference may be may multifactorial in nature. Our study was initiated at the same time with the use of the ultrasound-guided technique. As a result, the learning curve (gaining experience) for each surgeon that was participating in our protocol could potentially influence the development of complications. It is well-known that a significant learning curve exists in order to achieve an appropriate level of experience with adequate skills. As the study period progressed, the number of ultrasound-guided insertions of central venous catheters gradually increased, in comparison with the insertions that were performed via the open surgical technique. Apart from that, the incidence of failure that was associated with this technique was significantly decreased, along with the associated complications, a fact that was more pronounced toward the end of our study.

A recent survey that was based on children concluded that an ultrasound-guided supraclavicular approach of the brachiocephalic vein was recommended to reduce the number of attempts for cannulation and mechanical complications. Based on scarce publications on diagnostic and therapeutic strategies and on their experience (expert opinion), the panel proposed definitions and therapeutic strategies [20].

Furthermore, we would like to mention the results of another study, which attempted to evaluate the economic costs of cardiovascular diseases in Poland. They concluded that the estimated direct and indirect cost of cardiovascular diseases provide a useful input for economic impact assessments of public health programs and health technology analyses [21].

## 5. Conclusions

Based on the gained experience regarding the ultrasound-guided technique for the insertion of indwelling central venous catheters, it seems that the continuous and accurate imaging of the offending vessel along with the real-time monitoring of the tip of the catheter is mandatory in order to avoid complications. More precisely, this seems to be able to reduce the attempts to vessel catheterization, to eliminate the cases of inadvertent arterial puncture, as well as to reduce the incidence of injuries and scarring to the surrounding tissues. These, in turn, reduce the prevalence of stenosis, occlusion, and thrombosis of the lumen of the offending vessel. Moreover, it offers us the opportunity to re-insert a central venous catheter to the same recipient vein.

In cases where a repositioning is attempted, our suggestion is that an ultrasound imaging of the vessel should be performed, in order to exclude the possibility of venous thrombosis. This could be able to reduce the number of failure attempts to vessel catheterization, along with the risk of thromboembolism. The perioperative use of fluoroscopy in order to label the exact position of the tip of the wire (and, as a consequence, the tip of the catheter) increases significantly the possibility of successful catheter insertion and eliminates the inadvertent puncture of the carotid artery or of the anonymous vein. 

The most commonly utilized technique for the insertion of an indwelling central venous catheter in the pediatric patient population is the open surgical approach. This study revealed that the ultrasound-guided insertion technique is a safe alternative, is less time consuming, is devoid of serious complications, and protects the anatomic accuracy of the recipient vessel. It is used with an increasing frequency in the pediatric population of patients and is currently recommended as the procedure of choice for the insertion of indwelling central venous catheters.

## Figures and Tables

**Figure 1 diseases-11-00174-f001:**
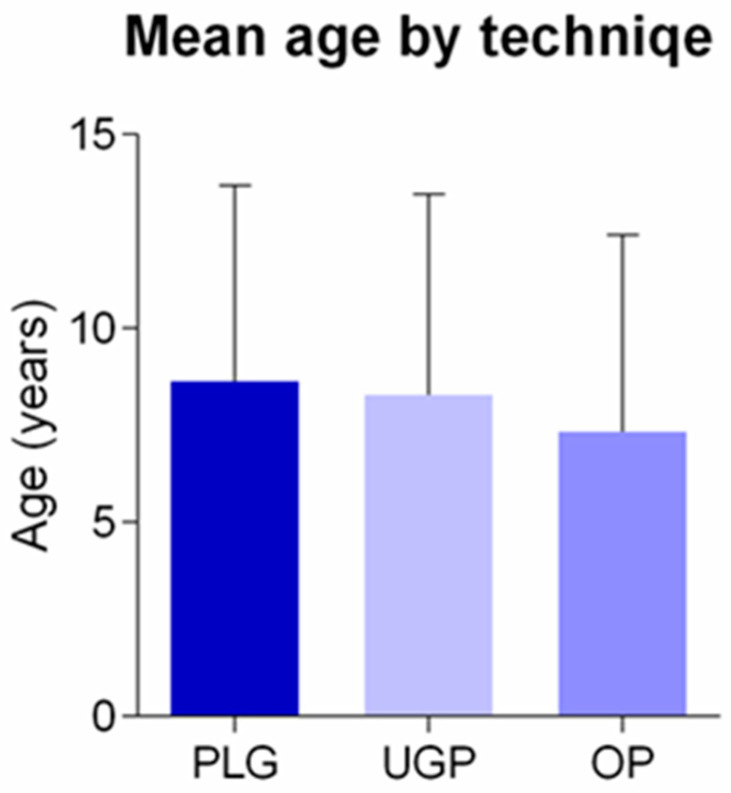
Figure depicting the mean age distribution of our participants, based on the adopted insertion technique.

**Figure 2 diseases-11-00174-f002:**
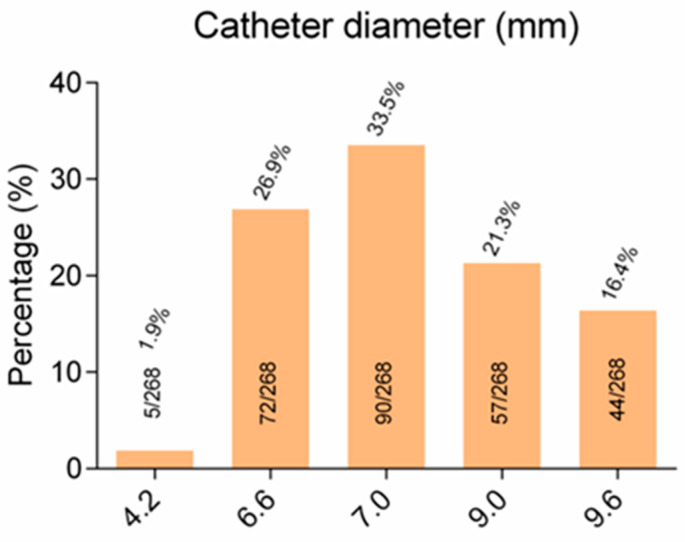
Figure depicting the distribution of internal central venous catheter diameter and their relative percentages.

**Figure 3 diseases-11-00174-f003:**
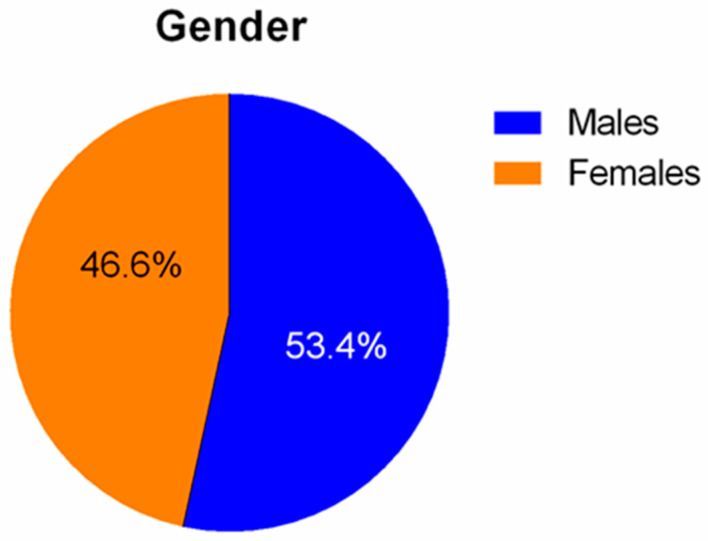
Schematic representation of the gender distribution of the participants to our survey.

**Figure 4 diseases-11-00174-f004:**
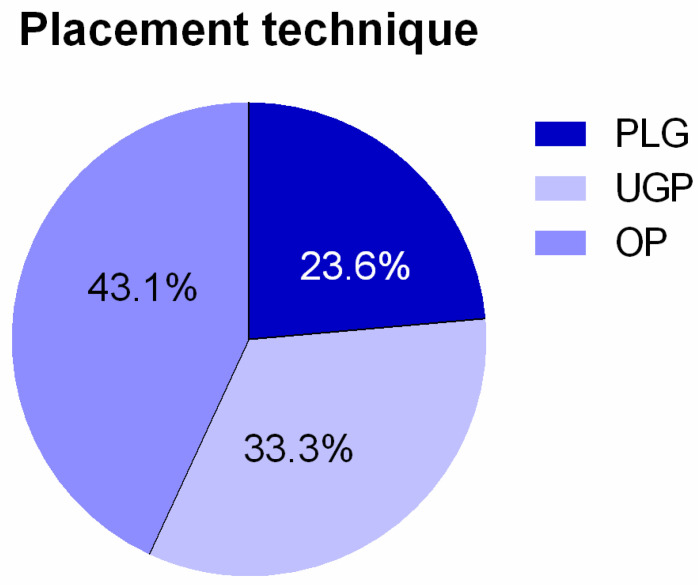
Figure representation, depicting the percentage of each placement technique for the central venous catheters.

**Figure 5 diseases-11-00174-f005:**
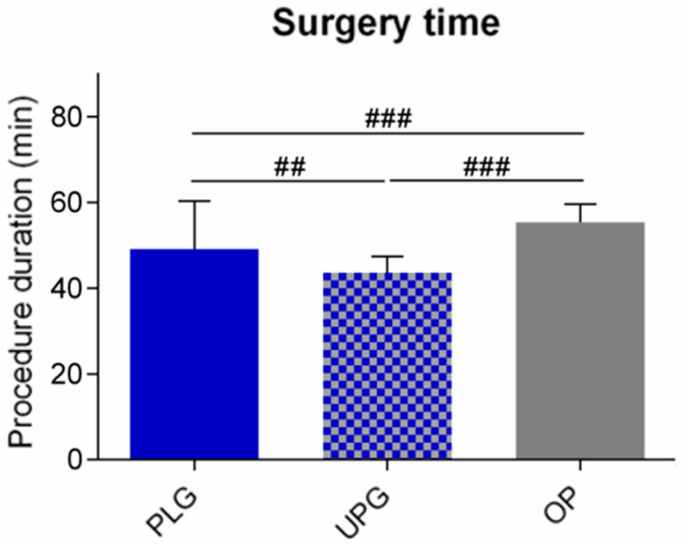
Schematic representation, depicting the mean operation time (procedure duration), based on the three different insertion techniques. More precisely, this schematic representation depicts the mean operation time (procedure duration), based on the three different insertion techniques. ### *p* < 0.001, ## *p* < 0.01 (Kruskal-Wallis with Dunn’s post hoc test).

**Figure 6 diseases-11-00174-f006:**
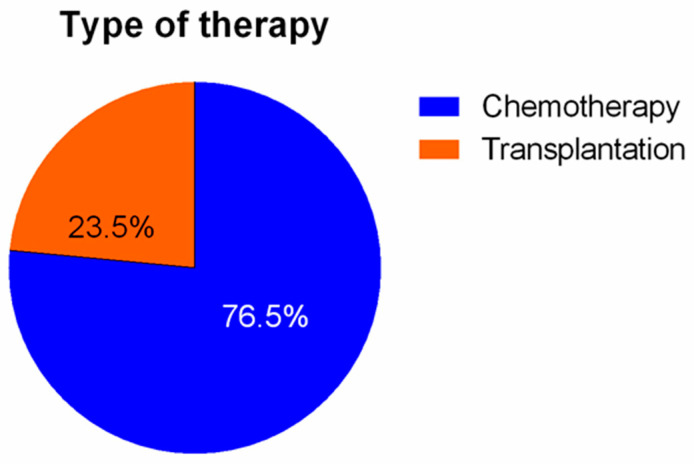
Figure representation, depicting the type of therapy (and their proportional participation in percent), according to our patient population.

**Figure 7 diseases-11-00174-f007:**
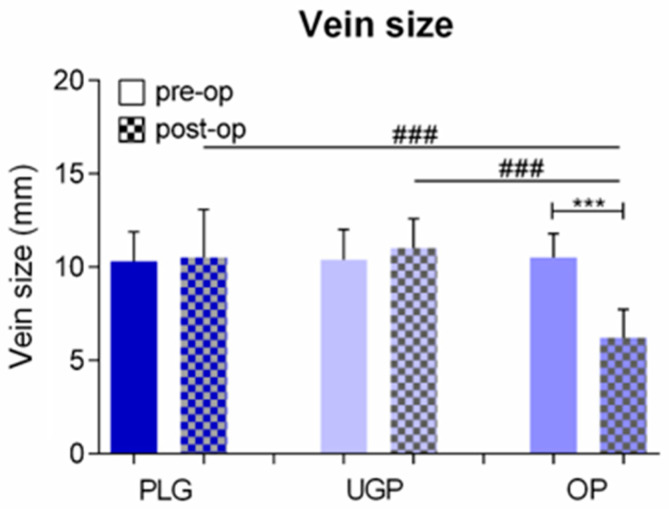
Schematic representation, depicting the size of the venous lumen before and after the insertion of the central catheter, referring to the three insertion techniques. More precisely, this schematic representation, depicts the size of the venous lumen before and after the insertion of the central catheter, referring to the three insertion techniques. ### *p* > 0.05 (Kruskal-Wallis followed by Dunn’s post hoc test); *** *p* < 0.05 (Student’s *t* test).

**Figure 8 diseases-11-00174-f008:**
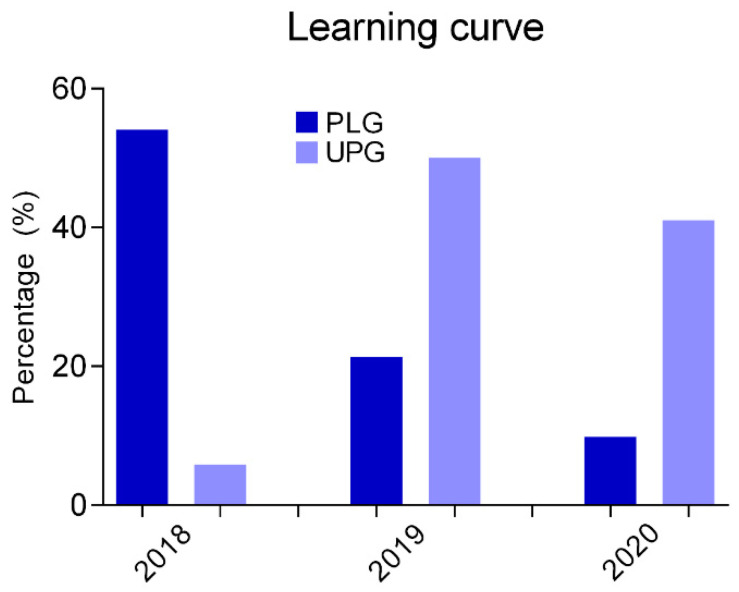
Schematic representation of the learning curve for each individual insertion technique, based on the data extracted from the participants to our study.

**Table 1 diseases-11-00174-t001:** General demographic characteristics of the participants, including their gender, mean patient age, insertion technique (PLG: Placement guided via external anatomical landmarks, UGP: ultrasound-guided placement, OP: open surgical insertion).

Patient Characteristics	n	%	*p*
Insertions (n)	268		
Gender (n)			
Male	142	53.4	*p* > 0.05
Female	126	46.6	
Mean age (years)	7.9 ± 5.1		
Insertion technique/Age			
PLG (n)	71	26.7	
age (years)	8.7 ± 5.1		*p* > 0.05
UGP (n)	87	33.2	
age (years)	8.0 ± 5.2		*p* > 0.05
OP (n)	110	40.1	
age (years)	7.3 ± 5.1		*p* > 0.05
Body Weight (kg)	29.3 ± 18.1		

**Table 2 diseases-11-00174-t002:** Categorization of patients based on their underlying pathological entity.

Disease	PLG	UGP	OP	Total
ALL	22 (35.48%)	3 (26.74%)	39 (35.45%)	84 (32.55%)
Sarcoma	10 (16.12%)	14 (16.27%)	15 (13.63%)	39 (15.11%)
Brain tumor	7 (11.29%)	5 (5.81%)	12 (10.90%)	24 (9.30%)
Lymphoma	2 (3.22%)	11 (12.79%)	10 (9.09%)	23 (8.91%)
AML	3 (4.83%)	10 (11.62%)	9 (8.18%)	22 (8.52%)
Solid tumor	4 (6.45%)	7 (8.13%)	10 (9.09%)	21 (8.13%)
Anemia	10 (16.12%)	9 (10.46%)	11 (10.00%)	30 (11.62%)
Immunodeficiency	2 (3.22%)	6 (6.97%)	4 (3.63%)	12 (4.65%)
Chest wall information	1 (1.61%)	1 (1.16%)	0 (0.00%)	2 (0.77%)
Histiocytosis	1 (1.61%)	0 (0.00%)	0 (0.00%)	1 (0.38%)
Total	62	86	110	258

**Table 3 diseases-11-00174-t003:** Detailed subdivision of patients, based on whether an initial insertion or re-insertion (successful or not) was attempted.

	n	%
Initial insertion	268/346	77.45
Reinsertion	78/346	19.9
Total number of insertions	346	100
Successful (Initial insertion	258/268	96.26
Unsuccessful (Initial insertion)	10/268	3.73
Total	268	100
Successful (Reinsertion)	69/78	88.46

**Table 4 diseases-11-00174-t004:** Categorization of central venous catheters based on their internal lumen diameter, as well as on the utilized insertion technique. A Fisher’s exact analysis was performed, which could not demonstrate a statistically significant difference regarding the number of the catheter’s lumens, when the three insertion techniques were compared (*p* = 0.4442).

Catheter Characteristics.	*n*	%
Diameter (French)		
4.2	5/268	1.9
6.6	72/268	26.9
7.0	90/268	33.5
9.0	57/268	21.3
9.6	44/268	16.4

**Table 5 diseases-11-00174-t005:** Single-Lumen (SL) and Double-Lumen (DL) catheters categorized based on insertion technique.

Lumens	SL	DL	Total
PLG	27 (38.02%)	44 (61.97%)	71 (100%)
UPG	40 (45.97%)	47 (54.02%)	87 (100%)
OP	52 (47.27%)	58 (52.72%)	110 (100%)
Total	119 (44.40%)	149 (55.60%)	268 (100%)

**Table 6 diseases-11-00174-t006:** Subdivision of patients, based on the utilized insertion technique.

Insertion Technique	*n*	*%*	*n*	%	*n*	%
PLG	88/346	25.4	71/268	26.49	62/258	23.6
UPG	121/346	34.9	87/268	32.46	86/258	33.3
OP	137/346	39.5	110/268	41.04	110/258	43.1
Total	346	100	268	100	258	100

**Table 7 diseases-11-00174-t007:** Subdivision of patients to different groups, based on the selected insertion technique for each individual patient subgroup. Patients are divided into subgroups based on their age distribution. Fisher’s exact analysis was insufficient to document a statistically significant difference between the three insertion techniques, based on the age distribution of the participants (*p* = 0.2864).

Technique/ Age Group	PLG [*n* (%)]	UPG [*n* (%)]	OP [*n* (%)]	Total [*n* (%)]
0–1	2 (3.22)	6 (6.97)	5 (4.54)	13 (5.03)
1–5	15 (24.19)	22 (25.58)	43 (39.09)	80 (31.00)
5–12	28 (45.16)	33 (38.37)	36 (32.73)	97 (37.59)
>12	17 (27.42)	25 (29.06)	26 (23.64)	68 (26.35)
Total	62	86	110	258

**Table 8 diseases-11-00174-t008:** Detailed presentation of the immediate complications following the insertion of central venous catheters, as they are distributed based on the insertion technique.

Immediately Complications	Insertion Technique			
	PLG [*n* (%)]	UGP [*n* (%)]	OP [*n* (%)]	Total [*n* (%)]
Rupture of carotid art	8 (12.90%)	7 (8.13%)	0 (0.00%)	15 (5.81%)
Suture vessel ligation	0 (0.00%)	0 (0.00%)	1 (0.99%)	1 (0.38%)
Inappropriate position	2 (3.23%)	3 (3.48%)	9 (8.18%)	14 (5.42%)
Hemopericardium	0 (0.00%)	1 (1.16%)	0 (0.00%)	1 (0.38%)
Pseudoaneurysm	1 (1.61%)	0 (0.00%)	0 (0.00%)	1 (0.38%)
Pneumothorax	2 (3.22%)	0 (0.00%)	0 (0.00%)	2 (0.77%)
Hemothorax	2 (3.22%)	0 (0.00%)	1 (0.90%)	3 (1.16%)
Hemorrhage	0 (0.00%)	0 (0.00%)	2 (1.81%)	2 (0.77%)
Total	15 (24.19%)	11 (12.79%)	13 (11.81%)	24 (9.30%)
	62	86	110	258
	(1)			

**Table 9 diseases-11-00174-t009:** Detailed presentation of all cases of inappropriate position of the distal end of the catheter, based on the anatomical location of the catheter tip.

Inappropriate Position	*n*	%
Total	14/258	5.42
Longer length	1/14	7.14
Azygos vein	1/14	7.14
Short length	4/14	28.57
R subclavian vein	6/14	42.85
Mediastinum	1/14	7.14
L Internal jugular vein	1/14	7.14

**Table 10 diseases-11-00174-t010:** Presentation of all mechanical complications, as well as of all long-term complications, implicated with central venous catheter insertion.

Mechanical Complications	PLG [*n* (%)]	UGP [*n* (%)]	OP [*n* (%)]	Total [*n* (%)]
Accidental remove	0 (0.00%)	2 (3.89%)	7 (7.60%)	9 (4.40%)
Catheter rupture	1 (1.72%)	2 (2.59%)	3 (3.26%)	6 (2.64%)

**Table 11 diseases-11-00174-t011:** Non-elective catheter removal.

Long-Term Complications	PLG [*n* (%)]	UGP [*n* (%)]	OP [*n* (%)]	Total [*n* (%)]
Infections	11 (18.96%)	7 (9.09%)	14 (15.21%)	32 (14.10%)
Malfunction	4 (6.89%)	4 (5.19%)	5 (5.43%)	13 (5.72%)
Thrombosis	5 (8.62%)	0 (0.00%)	7 (7.60%)	12 (5.28%)

**Table 12 diseases-11-00174-t012:** Presentation of the pre- and post- insertion internal diameter of the recipient vein, according to the utilized technique.

Technique	Vein Diameter (mm)	*p*
PLG (pre)	10.3	*p* > 0.05
PLG (post)	10.5	
UGP (pre)	10.4	*p* > 0.05
UGP (post)	11	
OP (pre)	10.6	*p* < 0.05
OP (post)	6	

**Table 13 diseases-11-00174-t013:** Presentation of the total number of cases of catheter removal, because of a complication of the therapeutic regimen, and their distribution along the different insertion techniques.

Removed	*n*	%	*n*	%
Total	227	100		
Complications of therapeutic regimen	163/227	71.80		
PLG	41/163	25.15	41/58	70.68
UGP	61/163	37.42	61/77	79.22
OP	61/163	37.42	61/92	66.30

**Table 14 diseases-11-00174-t014:** Presentation of the total length of stay of indwelling central venous catheters. Patients are subdivided into several groups according to the insertion technique.

Average Length of Stay	Days ± SD	*p*
Total	236.0 ± 230.9	*p* > 0.05
PLG	238.3 ± 235.7	
UGP	242.2 ± 233.8	
OP	229.5 ± 227.3	

## Data Availability

No new data were created.

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
