# Peer review of "Open versus Transcutaneous (Ultrasound-Guided and Based on Anatomic Landmarks) Tunneled Venous Access to the Right Internal Jugular Vein in Children: A Prospective Single-Center Study"

_diseases, 2023, doi:10.3390/diseases11040174_

Round 1
Reviewer 1 Report
Comments and Suggestions for Authors
This manuscript offers valuable insights into the effective treatment of pediatric patients, focusing on three primary methods of venous catheterization. The prospective study compares the immediate and long-term outcomes of ultrasound-guided, transcutaneous, and open surgical techniques involving the right jugular vein in children.
The abstract succinctly encapsulates the comprehensive content of the lengthy write-up, serving as an excellent summary. The introduction provides a solid foundation for the study, presenting the latest information on the three catheterization methods. Despite the potential existence of anecdotal data predicting outcomes, the study is well-justified and provides empirical evidence supporting the increasing application of the ultrasound-guided technique.
The method section is well-crafted, emphasizing exclusion criteria and reasons for inclusion or exclusion, while also addressing ethical considerations. The pros and cons of each method are thoroughly documented across materials and methods, results, and discussion sections. However, there is noticeable repetitiveness throughout the manuscript, including figures, tables, and schemes presenting similar data in various forms. The prevalence of univariate analysis in the results section could be balanced with more multivariate analysis, reducing the number of figures, tables, and diagrams. Organizing the table results in a ranked order would enhance understanding.
The discussion section, although detailed and drawing on other publications to clarify study results, echoes some information from the results section, potentially causing reader disinterest. This paper holds significant relevance for clinicians, prompting potential questions and further research opportunities.
In conclusion, with these minor modifications, I recommend this manuscript for publication due to its valuable contribution to the field.
Comments on the Quality of English Language
The proficiency in English is commendable and meets the required standards. Nevertheless, it is advisable to summarize the results and discussion sections in order to enhance clarity and coherence.
Minor editorial and grammatical errors, especially in lines 193-194, 201, and 664, need attention. Consistency in using commas or periods for numerical data and rates should be ensured throughout the paper.
Minor editorial and grammatical errors, especially in lines 193-194, 201, and 664, need attention. Consistency in using commas or periods for numerical data and rates should be ensured throughout the paper.
Author Response
Dear Reviewer,
Thank you for for valuable comments, I strongly consider that they enhance the validity of our study.Regarding your review, I would like to state the following:
You have mentioned that' The pros and cons of each method are thoroughly documented across materials and methods, results, and discussion sections. However, there is noticeable repetitiveness throughout the manuscript, including figures, tables, and schemes presenting similar data in various forms. The prevalence of univariate analysis in the results section could be balanced with more multivariate analysis, reducing the number of figures, tables, and diagrams. Organizing the table results in a ranked order would enhance understanding.'
I strongly appreciate your comments. Some tables have been removed from the revised version of our manuscript, as they seem to present similar data in various forms.
You have also mentioned that ' The discussion section, although detailed and drawing on other publications to clarify study results, echoes some information from the results section, potentially causing reader disinterest.'
Based on your comment, several contents of the discussion section have been removed as they were already presented in the results section.
Reviewer 2 Report
Comments and Suggestions for Authors
This prospective, single-center study compared the immediate and long-term complications associated with the utilized techniques for the insertion of indwelling central venous catheters, mainly focused on the open surgical technique, the ultrasound-guided technique and the transcutaneous technique based on external anatomical landmarks in the right internal jugular vein in a pediatric population. Here are some concerns about this study:
1. Please describe the detailed procedure, including the open surgical technique, the ultrasound-guided technique, and the transcutaneous technique based on external anatomical landmarks.
2. The patients were divided to three groups, please list the include and exclude criteria for each group.
3. Is the open surgical technique conducted because of the failure of the other two procedures? If so, then the open surgical technique group should be considered as a subgroup of Group A and Group B, instead of being Group C.
4. The quality of tables and figures can be improved. The tables and figures would be more organized and focused. Moreover, the fonts from different tables were different. Please double check.
Comments on the Quality of English LanguageN/A
Author Response
Dear Reviewer,
I really appreciate your valuable comments as they can significantly enhance the scientific soundness of our study. Regarding your remarks, I would like to state the following:
You have mentioned that'
Please describe the detailed procedure, including the open surgical technique, the ultrasound-guided technique, and the transcutaneous technique based on external anatomical landmarks.
2. The patients were divided to three groups, please list the include and exclude criteria for each group.
3. Is the open surgical technique conducted because of the failure of the other two procedures? If so, then the open surgical technique group should be considered as a subgroup of Group A and Group B, instead of being Group C.'
I strongly appreciate your valuable comments. All necessary declarations and corrections are added to the revised version of our manuscript.
Moreover, you have noticed that ' The quality of tables and figures can be improved. The tables and figures would be more organized and focused. Moreover, the fonts from different tables were different. Please double check.'
Based on your valuable comments, several modifications are added to the revised version of our manuscript, as requested.
Reviewer 3 Report
Comments and Suggestions for Authors
The article by Kouna et al. "Open versus transcutaneous (ultrasound-guided and based on anatomic landmarks) tunneled venous access to the right internal jugular vein in children: A prospective single-center study. covers a potentially interesting and emerging topic related to the paediatric anesthesiology. In this sense, this remains to be potentially interesting for thediseases readers. I regard the main point of this paper as highly attractive as well as the results are clearly presented. The text does not contain any major errors, therefore I have some minor comments and recommendations:
1. There is a need to provide slightly more expanded introduction shortly mentioning/describing anatomical explanation and children adults differences.
2. The figure in introduction section should be added.
3. Following references should be added and properly cited within the main text:
- Timsit JF, Baleine J, Bernard L, Calvino-Gunther S, Darmon M, Dellamonica J, Desruennes E, Leone M, Lepape A, Leroy O, Lucet JC, Merchaoui Z, Mimoz O, Misset B, Parienti JJ, Quenot JP, Roch A, Schmidt M, Slama M, Souweine B, Zahar JR, Zingg W, Bodet-Contentin L, Maxime V. Expert consensus-based clinical practice guidelines management of intravascular catheters in the intensive care unit. Ann Intensive Care. 2020 Sep 7;10(1):118. doi: 10.1186/s13613-020-00713-4.
- Mela A, Rdzanek E, Poniatowski ŁA, Jaroszyński J, Furtak-Niczyporuk M, Gałązka-Sobotka M, Olejniczak D, Niewada M, Staniszewska A. Economic Costs of Cardiovascular Diseases in Poland Estimates for 2015-2017 Years. Front Pharmacol. 2020 Sep 8;11:1231. doi: 10.3389/fphar.2020.01231.
4. In some places the use of English could be improved on.
Completing this gaps will have an impact on the understanding the aim of the study and, from my point of view, is absolutely necessary.
Comments on the Quality of English LanguageMinor review
Author Response
Dear Reviewer,
I strongly appreciate your valuable comments as they increase the overall merit of our study for the readers.
Regarding your recommendations, I would like to state the following:
You have mentioned that'
There is a need to provide slightly more expanded introduction shortly mentioning/describing anatomical explanation and children adults differences.
- The figure in introduction section should be added.' I strongly appreciate your comments. All relevant corrections are included in the revised version of our manuscript, as requested. Moreover, you have recommended that'
Following references should be added and properly cited within the main text:
- Timsit JF, Baleine J, Bernard L, Calvino-Gunther S, Darmon M, Dellamonica J, Desruennes E, Leone M, Lepape A, Leroy O, Lucet JC, Merchaoui Z, Mimoz O, Misset B, Parienti JJ, Quenot JP, Roch A, Schmidt M, Slama M, Souweine B, Zahar JR, Zingg W, Bodet-Contentin L, Maxime V. Expert consensus-based clinical practice guidelines management of intravascular catheters in the intensive care unit. Ann Intensive Care. 2020 Sep 7;10(1):118. doi: 10.1186/s13613-020-00713-4.
Mela A, Rdzanek E, Poniatowski ŁA, Jaroszyński J, Furtak-Niczyporuk M, Gałązka-Sobotka M, Olejniczak D, Niewada M, Staniszewska A. Economic Costs of Cardiovascular Diseases in Poland Estimates for 2015-2017 Years. Front Pharmacol. 2020 Sep 8;11:1231. doi: 10.3389/fphar.2020.01231.'
I have included these references in the discussion section of our manuscript, along with a seperate section, analyzing their conclusions. Finally, you have suggested that' In some places the use of English could be improved on.' I strongly appreciate your comment and some language corrections are included in the revised section of our manuscript.
Round 2
Reviewer 2 Report
Comments and Suggestions for Authors
The authors addressed all the comments and suggestions. But please double-check Table 1~3 vs Table 4~12. The font in the table is different.